# Sociality of Cats toward Humans Can Be Influenced by Hormonal and Socio-Environmental Factors: Pilot Study

**DOI:** 10.3390/ani13010146

**Published:** 2022-12-30

**Authors:** Hikari Koyasu, Hironobu Takahashi, Ikuto Sasao, Saho Takagi, Miho Nagasawa, Takefumi Kikusui

**Affiliations:** 1Laboratory of Human-Animal Interaction and Reciprocity, Azabu University, Sagamihara 252-5201, Japan; 2Japan Society for the Promotion of Science, Tokyo 102-8471, Japan

**Keywords:** cat behavior, animal socialization, cat-human interaction, testosterone, cortisol, oxytocin

## Abstract

**Simple Summary:**

Cats are the most widely kept companion animal in the world. Various factors influence the sociality of cats. Here, we investigated whether the hormonal status of cats, and the age at which they began living with a human, affected their behaviors toward humans. The results showed that male cats that began living with a human earlier had more contact with humans. In addition, males with lower testosterone levels had more contact with humans. The results of this pilot study suggest that testosterone levels and the timing of when cats begin living with humans modulate affinity behavior of male cats toward humans.

**Abstract:**

Individual differences in the sociality of cats are influenced by inherited and environmental factors. We recently revealed that hormones can make a difference in intraspecies social behavior. It remains unclear whether cat behavior toward humans is modulated by hormones. Therefore, we analyzed the relationship between cat behavior and their basal hormone concentrations after spending time together with human experimenters. In addition, we analyzed the relationship between cat behavior and the timing of when the individual cats began living with a human because the sociality of cats could be dependent on their developmental experiences. The results showed that male cats that began living with humans earlier had more contact with an experimenter. In addition, individual male cats with low testosterone levels were more likely to interact with an experimenter. These findings of this pilot study suggest that the sociality of male cats toward humans is affected by testosterone and the age at which they begin to live with humans.

## 1. Introduction

Animals live in relationships with other members of their species, as well as with different species. Cats, in particular, live in the same space with humans and have a close relationship through their interactions based on various senses [1]. Cats have many natural instincts that promote human interaction [1]. In vocal communication, for example, meowing has changed to communicate more effectively with humans. Although meowing is typically only used by mothers and kittens among cats [2], they also meow to attract human attention, even after they become adult animals [3]. Visual communication has also developed between cats and humans [1]. Cats follow human gaze in food selection tasks for referential information [4], and eyeblink synchronization, an indicator of smooth communication, has also been observed [5,6,7]. 

Although cats have evolved various behaviors toward humans, there is large individual variability. Although cats have evolved various behaviors toward humans, there is large individual variability owing to selection preference that favors appearance over sociality [8]. These differences in behavior are influenced by genetic and environmental factors. The question of whether genetic factors influence animal behavior has been analyzed by investigating behavioral differences between breeds, and supported by the responses to questionnaires [9,10]. In addition, coat color, which is genetically determined, has been studied in relationship to behavior in cats [11,12,13]. 

Furthermore, genetic variation can lead to differences in the development of the hormone system in cats and the amount of basal hormone secretion, which could influence cat behavior. Another factor that influences cat behavior toward humans, such as early childhood experiences [14,15,16,17,18,19,20]. Cats receiving 30–40 min of handling per day when they are kittens display a greater affinity towards humans [16]. In addition, the number of handlers that kittens are exposed to affects their affinity towards humans as adult cats [16,17]. We previously showed that intraspecific behavior in cats is modulated by hormones, and it is possible that their behavior toward humans is also modulated by hormones. Carlstead et al. showed that cat urinary cortisol is negatively correlated with hiding from human behaviors. In addition, concentrations of urinary cortisol [21] and oxytocin [22] have been shown to change depending on how humans care for cats. Finally, aggressive behavior decreases in cats after castration [23,24], which may be due to decreased testosterone concentration.

Although this was a pilot study due to the limited number of sample size, we aimed to clarify whether the behavior of cats toward humans is influenced by the age at which cats begin living with humans and/or by cat hormone levels. We recorded the age at which the cats began to live with humans, their behavior while spending time with human experimenters, and measured their hormone levels. We hypothesized that (1) the younger a cat begins to live with humans, the greater the affinity they would have with humans as adult cats, (2) individual cats with lower testosterone concentrations would avoid experimenters less than that of cats with higher testosterone concentrations, (3) individual cats with lower cortisol concentrations would avoid experimenters less than that of cats with higher cortisol concentrations, and (4) individual cats with lower oxytocin concentrations would exhibit lower group boundaries, interact with others, and develop more friendly interactions with experimenters.

## 2. Materials and Methods

### 2.1. Cat Subjects

Cats living in a cat café or shelter participated in this experiment, and all of the cats were rescued cats looking for adoptive homes. There were 4 male and 11 female cats aged 20.0 ± 12.6 months of age; all of the male cats were castrated. Detailed information about each cat is shown in Appendix A.

### 2.2. Experimental Environment

For the café cats, the experiment was conducted in an 8.6 m^2^ room of the cat café. Because the shelter cats had been kept at Azabu University for 3 months for another experiment, we used a 10 m^2^ room at the Companion Dog Laboratory of Azabu University as the experimental room for the shelter cats. Within each room, there was a sofa for the human experimenters and a cat bed or cat tower for cats to choose where they wanted to rest. The environment of the two rooms was almost the same and both rooms were new places for the cats.

### 2.3. Behavioral Observation Flow-Chart

We investigated the behavior of the cats toward the experimenter over a period of 2 h. A flow-chart of the experiment is shown in Figure 1. First, the cat was introduced to the experimental room with the experimenter petting and talking to the cat for 10–20 min to get used to the room. Second, after the cat walked freely and explored the room, the experimenter left the room and the cat spent 5 min alone. Third, the experimenter re-entered the room and spent 2 h with the cat. During this time, the experimenter read a book or worked on a computer and was not allowed any interaction with the cat, such as petting, talking to, or looking at the cat. The experimenters included two males and two females who had met the subject cat at least five times, but less than ten times. One recording camera (HDR-AS50R, SONY) was installed on the ceiling of each room and a second camera was used in the case of blind areas.

### 2.4. Behavioral Analysis

The behavior of cats toward experimenters was recorded. The following behaviors were analyzed: rubbing against experimenter, being on the same sofa with experimenter, touching, meowing, sniffing, staying near the door, greeting score, and following score. Greeting and following were scored based on the Strange Situation Test [25]. The following score was calculated when the experimenter left the room once after the familiarization period, and the greeting score was calculated when the experimenter re-entered the room. Other behaviors during the reunion were recorded by continuous sampling. The definition of each behavior is listed in Appendix A.

### 2.5. Hormone Assay

Cat urine samples were collected with cotton immediately after urination using a two-layer toilet between 7:00 a.m. and 10:00 a.m. A total of 101 urine samples (6.7 ± 3.6 samples/cat) were collected, and the experimenter was able to determine which cat excreted the urine in each case. Urine samples were collected during the month before and after the experiment. Basal hormone concentrations for each individual cat were obtained by averaging samples.

#### 2.5.1. Cortisol Concentrations

Cortisol concentrations were measured using an enzyme-linked immunosorbent assay (ELISA). We prepared the ELISA-plate using a mouse IgG-Fc fragment antibody (A90-131A, Bethyl Laboratories, Montgomery, TX, USA). The secondary antibody was diluted 500-fold and dispensed in 100 µL aliquots. After overnight incubation at 22–25 °C, the liquid in each well was discarded. Subsequently, a phosphate buffer containing 0.1% bovine serum albumin (BSA) was dispensed in 200 µL aliquots to all wells. After incubation for 30 min at 22–25 °C, the plates were stored at 4 °C in the dark until used in the assay.

Undiluted urine samples were dispensed into the wells of the ELISA-plate. An anti-cortisol antibody (ab1949; Abcam, Cambridge, UK) diluted 200,000-fold was used as a primary antibody. Cortisol-3-CMO-HRP (FKA403, COSMO, Dublin, Ireland) diluted 10,000-fold was used as a horseradish peroxidase (HRP). 15 µL standard samples, 15 µL urine samples and 100 µL primary and 100 µL secondary antibodies were dispensed into each well. After incubation for at least 6 h, the liquid in each well was discarded and the plate was washed four times using a plate washer. Thereafter, 150 μL of substrate buffer was dispensed into all wells. The reaction was stopped by adding 50 μL of 4 N H_2_SO_4_. Absorbance was measured at a wavelength of 450 nm using a microplate reader (Model 680XR, Bio-Rad Laboratories, Inc., Hercules, CA, USA).

#### 2.5.2. Testosterone Concentrations

Testosterone concentrations were also measured using ELISA and the same plates as used for cortisol assay. After washing the plate three times, urine samples were dispensed into the wells. Urine samples with higher concentrations were diluted 2-fold with a phosphate buffer containing 0.1% BSA before dispensing. An anti-testosterone 3 CMO antibody (ab35878, Abcam) diluted 25,000-fold was used as a primary antibody, and a mouse IgG-Fc fragment antibody (A90-131A, Bethyl Laboratories) diluted 500-fold was used as secondary antibody. The HRP used in this process was Testosterone-3-CMO-HRP (FKA101, COSMO) that was diluted 10,000-fold. 25 µL standard samples and 25 µL urine samples were dispensed in each well. Then, 100 µL primary antibody, 100 µL secondary antibody and 100 µL HRP were dispensed. After incubation for at least 6 h, the liquid in the plate was discarded and the plates were washed four times using a plate washer. Thereafter, 150 μL of substrate buffer was dispensed into all wells. The reaction was stopped by adding 50 μL of 4 N H_2_SO_4_. Absorbance was measured at a wavelength of 450 nm using a microplate reader. 

#### 2.5.3. Oxytocin Concentrations

We used a commercially available oxytocin ELISA kit (ADI-901-153A-0001, ENZO, New York, NY, USA) for the assay. Urine samples diluted 50-fold with the assay buffer in the kit were dispensed into the wells of the ELISA-plates. 15 µL standard samples and 15 µL urine samples were dispensed in each well. A primary antibody and HRP were dispensed in 50 µL aliquots in each well. After incubation at 4 °C for 18–24 h, the substrate solution was dispensed in 200 µL aliquots in all wells. After incubation at 22–25 °C for 1 h, 50 μL of stop solution was added to each well. Absorbance was measured at a wavelength of 405 nm using a microplate reader.

#### 2.5.4. Creatinine Concentration

A creatinine standard samples and the urine samples diluted 100-fold with distilled water were dispensed into a 96-well microplate (AS ONE Co., Ltd., Osaka, Japan) at 100 μL each, followed by 50 μL of 1 M NaOH and 50 μL of 1 g/dL trinitrophenol. The absorbance was measured at a wavelength of 490 nm using a microplate reader after the plate was left at room temperature (22–25 °C) for 20 min. The samples were used undiluted. To adjust for variations in urine concentration, all hormone concentrations were indexed against the creatinine concentration.

### 2.6. Statistical Analysis

All statistical analyses were conducted using R version 3.5.1. Linear regression model and correlation analysis were conducted to investigate the relationship between cat behavior toward humans and their age in months and hormone concentrations. Each social behavior towards humans was analyzed using a linear model (LM) and the lmer function in the lme4 package version 1.1.10. Testosterone concentration, cortisol concentration, oxytocin concentration, age at experiment, and age at which cats began to live with humans were entered as fixed factors. Correlation analysis was conducted separately for males and females since an analysis including interactions between sex and other factors could not be performed due to the lack of samples. In the correlation analysis, spearman’s rank correlation coefficient was used. The significance level was adjusted because the analysis was repeated five times with months and hormones for a behavior (significance was set at *p* < 0.01).

## 3. Results

### 3.1. Effects of Age and Hormones on Cats’ Behaviors toward Humans

None of the behaviors examined was affected by age or hormone concentrations (Appendix A).

### 3.2. Correlation of Male Cats’ Behavior toward Humans with Age and Hormones

We then examined correlations between behavior, age, and hormone concentrations separately by sex. As a result, the age at which male cats began living with humans correlated negatively with sharing sofa (Figure 2A; *rs* = −1.000, *p* < 0.001), contact (Figure 2B; *rs* = −1.000, *p* < 0.001), rubbing (Figure 2C; *rs* = −1.000, *p* < 0.001), tail up (Figure 2D; *rs* = −1.000, *p* < 0.001), and following (Figure 2E; *rs* = −1.000, *p* < 0.001). In males, testosterone concentrations correlated negatively with sharing sofa (Figure 3A; *rs* = −1.000, *p* < 0.001), contact (Figure 3B; *rs* = −1.000, *p* < 0.001), rubbing (Figure 3C; *rs* = −1.000, *p* < 0.001). In addition, there was a positive correlation between the age at which the cats began living with humans and testosterone concentrations (Appendix A; *rs* = 1.000, *p* < 0.001). The results of the correlation between all behaviors, hormones, and age in months are shown in Appendix A.

### 3.3. Correlation of Female Cats’ Behavior toward Humans with Age and Hormones

In females, there was no correlation between behaviors towards humans and age or hormone concentrations.

## 4. Discussion

In this study, we investigated the relationship between the behavior of cats toward humans and their age and hormone concentrations to determine whether behavior toward humans is influenced by the age at which cats begin living with humans and/or their hormone levels. We found that the earlier a male cat began living with humans and the lower the testosterone level, the more likely the cats were to interact with experimenters. Only the results in males were consistent with our original hypothesis.

A decrease in testosterone leads to a decrease in aggression. The relationship has been reported in various species, including rodents, e.g., [26,27,28], primates, e.g., [29,30,31,32], and canidae, e.g., [33]. Lower testosterone levels may have made the cats more tolerant of other individual cats and increased their contact with humans. This association was only observed in males, which may have been influenced by sexual differentiation of the brain. For example, the brain of rats is sexually undifferentiated until about one week after birth. Androgen action during the perinatal period results in masculinization and defeminization of their brain [34,35,36,37,38]. Individuals that develop masculine brains during this period depend on testosterone for their behavior even after growth [39]. Additionally, in cats, the sexual differentiation of the brain during fetal life might have resulted in different outcomes for males and females. In addition, androgen precursors are also secreted from the adrenal glands in both sexes and converted to testosterone in the peripheral tissues [40]. Although all of the male cats were castrated, the testosterone produced by their adrenal glands could have affected their behavior. In addition, it has been reported that neutered males are friendlier to humans than neutered female [41]. The sex differences in hormone-behavior associations shown in this study may help us understand sex differences in behavior toward humans.

Individual male cats that began to live with a human later in life spent more time at the door, avoiding humans, which is consistent with previous studies. Early handling has shown positive effects on behavior in several studies of cat behavior towards humans [16,18,19,20]. The results of this study support these studies that the timing of first human contact during development is an important factor in a cat’s socialization with humans. There is another possibility that the cats beginning to live with humans early in their lives had lower testosterone baselines, which may have affected the cats’ behavior toward humans, since there was also a relationship between the age at which they began to live with humans and their testosterone levels.

Since cortisol and oxytocin are related to the behavior of cats toward other cats [42,43], we predicted that these hormones would also affect the behavior of cats toward humans. Our results did not support the prediction. Cortisol is also associated with anxiety and aggression in cats [43,44], but none of the individuals in this study showed excessively anxiety or aggression toward humans. In a previous study examining the relationship between fear responses to humans and cortisol, there was no relationship between them [45]. These may be the reasons why there was no correlation between cortisol and social behavior. In addition, oxytocin acts on a specific individual, especially the owner, and its effects depend on whom it interacts with. In dogs, oxytocin is increased by interaction with owners and familiar humans [46,47,48,49,50,51]. In this experiment, the experimenter met more than five times with each individual cat. It is possible that the effect of oxytocin is not present in these experimenters or in the entire category of humans. In addition, the first human encounter for cats is important for the subsequent relationship because it makes clear the other’s motivations and behavioral strategies. Because some differences in cat behavior occur based on the attributes of humans [3], investigating the behavior of cats toward a variety of humans, including strangers and familiar people, will lead to further understanding of cat social behaviors.

Most rarely a single hormone influences behavior; multiple hormones influence behavior integratedly. For example, testosterone secretion in the testis is affected by cortisol in an in vitro study of cats [52]. Other hormones, such as serotonin, also influence cat behavior [53], but the relationships between serotonin, oxytocin and testosterone are unknown.

In previous study, sociality with humans can be explained by paternal inheritance [54,55]. McCune examined how childhood experiences of interaction with humans influence later behavior, and whether these influences are affected by paternity [53]. As a result, individual cats that had a father who was friendly to humans, and experienced human contact from an early age, approached humans earlier and spent more time with humans than those who had not experienced human contact or had an unfriendly father [54]. These observations indicate that cat sociality toward humans is influenced by both environmental and inherited factors. Therefore, in the future, more profound insights into the sociality of cats with humans may be gained by examining the genetic background of individual cats and their behavior.

Although it has been shown that testosterone concentrations and the age at which they began living with humans may influence the behaviors of males toward humans, there are some limitations of this pilot study. One of the most concerns is the small sample size. In particular, only four males were sampled, which is not a sufficient number of data to account for all cats. Additional experiments need to be conducted to confirm reproducibility with more samples. Second, it is possible that the indicator of when cats began to live with humans is not appropriate. Although we used the age in months when the cats were rescued as an indicator of when they began interacting with humans, they might have been interacting with humans even before they were rescued. It will be necessary to conduct experiments on individual cats so that we can accurately determine when they start interacting with humans.

## 5. Conclusions

In this pilot study, there was no relationship between hormones or age and behaviors toward humans in male and female cats overall. In only male cats, the lower testosterone and the earlier they began to live with humans, the more interactions they had with humans. These results suggest one possibility that socialization of male cats with humans is influenced by both experience in kittenhood and testosterone, although reproducibility needs to be confirmed. Research on the relationship between cat behavior and endocrine status could be key to revealing the evolution of cat sociality.

## Figures and Tables

**Figure 1 animals-13-00146-f001:**
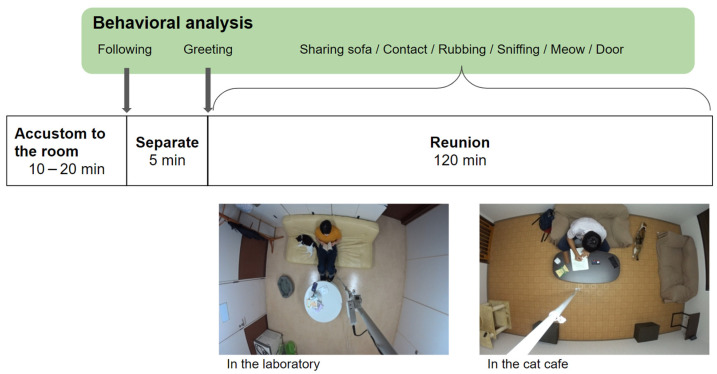
Experimental procedure.

**Figure 2 animals-13-00146-f002:**
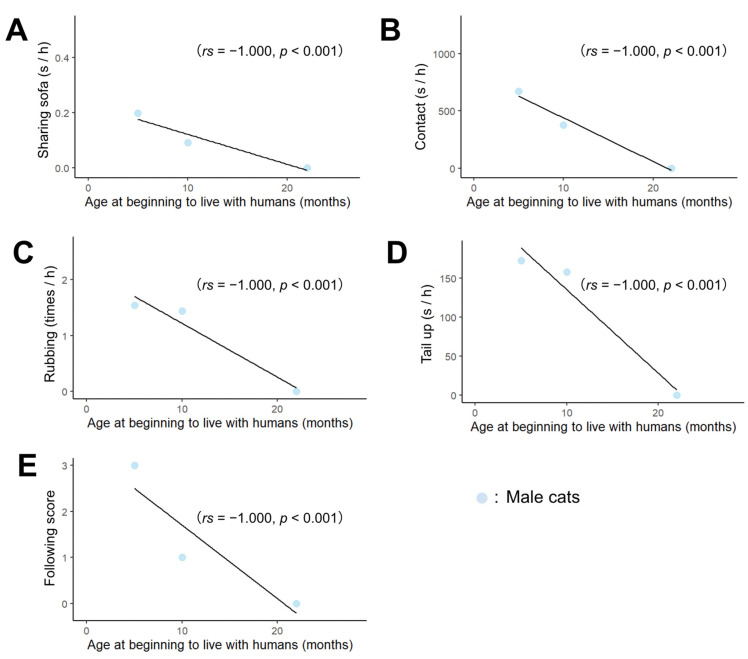
Correlation between behavior towards experimenters and age at keeping with humans (months). (**A**–**E**) indicate the relationship between age at beginning to live with humans and the following behaviors, (**A**) Sharing sofa, (**B**) Contact, (**C**) Rubbing, (**D**) Tail up, (**E**) Following.

**Figure 3 animals-13-00146-f003:**
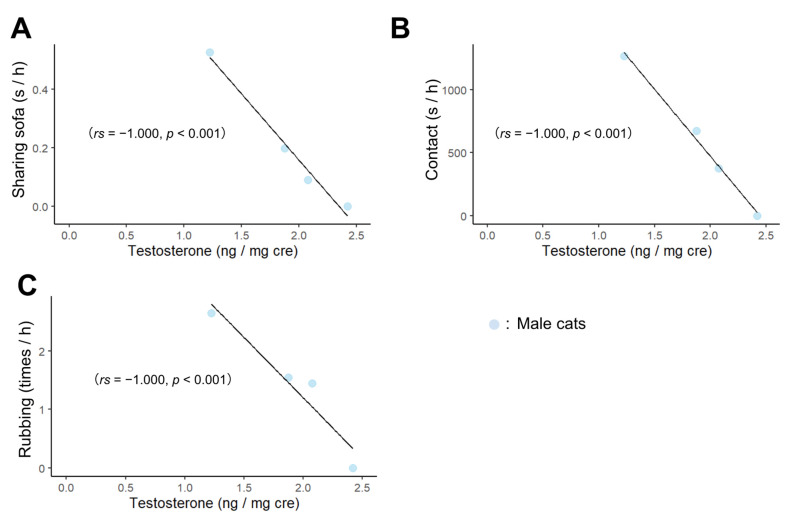
Correlation between behavior towards experimenters and testosterone concentrations. (**A**–**C**) indicate the relationship between testosterone concentration and the following behaviors, (**A**) Sharing sofa, (**B**) Contact, (**C**) Rubbing.

## Data Availability

The data presented in this study are available within the article and in the Appendix A.

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
