# Peer review of "Sociality of Cats toward Humans Can Be Influenced by Hormonal and Socio-Environmental Factors: Pilot Study"

_animals, 2022, doi:10.3390/ani13010146_

Round 1
Reviewer 1 Report
This study involves 4 male and 11 female cats, all neutered. All save one are two years or less of age—quite young. Especially with the hormonal emphasis, it would make sense if the data were more clearly split to display the male and female data separately. It was difficult to know whether graphs were based on all 15 cats or only the 4 males. Early handling has shown positive effects on behavior in several studies of cat behavior with humans—suggesting a sort of critical period (Karsh 1983; Bradshaw, 1992; Casey 2008; Turner, 2017; Hart, 2018). Males sometimes show up as more friendly than females. Young cats generally are more friendly than old ones. This paper is suggesting that flexibility in young male cats’ (but not female cats’) friendliness to humans hinges on early exposure to living with humans.
The highlighted results here are derived from only 4 male cats who were affected by the ages at which they first lived with humans and by their testosterone hormonal levels. It is concerning to make serious conclusions based on only 4 neutered male cats (10, 15, 24, 24 months of age). How can these results be justified based on only 4 males? If published, some of these issues need to be handled more transparently and addressed in a LIMITATIONS section.
Typos, should be none is, not none are
Author Response
This study involves 4 male and 11 female cats, all neutered. All save one are two years or less of age—quite young. Especially with the hormonal emphasis, it would make sense if the data were more clearly split to display the male and female data separately. It was difficult to know whether graphs were based on all 15 cats or only the 4 males. Early handling has shown positive effects on behavior in several studies of cat behavior with humans—suggesting a sort of critical period (Karsh 1983; Bradshaw, 1992; Casey 2008; Turner, 2017; Hart, 2018). Males sometimes show up as more friendly than females. Young cats generally are more friendly than old ones. This paper is suggesting that flexibility in young male cats’ (but not female cats’) friendliness to humans hinges on early exposure to living with humans.
>We appreciate the helpful comments and taking the time to review. Our responses to each comment can be found below.
The difficulty in understanding whether the results are for males and females or only males
We have added a legend to the figure 2 and 3 to make it easier to understand that the figure is the result of males. In addition, sub-headings were added to the main text to make it easier to understand that the results are for males and females, for males only, and for females only (line 187, 190, 200).
Adding some references
A few references were lacking, so we have added them in introduction and discussion.
・The effects of additional socialisation for kittens in a rescue centre on their behaviour and suitability as a pet, Casey RA, Bradshaw JW, Appl. Anim. Behav. Sci., 2008
・A review of over three decades of research on cat-human and human-cat interactions and relationships, Turner DC, Behav. Processes, 2017
The highlighted results here are derived from only 4 male cats who were affected by the ages at which they first lived with humans and by their testosterone hormonal levels. It is concerning to make serious conclusions based on only 4 neutered male cats (10, 15, 24, 24 months of age). How can these results be justified based on only 4 males? If published, some of these issues need to be handled more transparently and addressed in a LIMITATIONS section.
>We agree with the comments. The paragraph about limitations of this study has been added to make clear the small sample size and other concerns (line 277-287). In addition, we have positioned this paper as a pilot study.
Typos, should be none is, not none are
>We have corrected (line 188).
Reviewer 2 Report
Comments on the manuscript “Sociality of cats toward humans can be influenced by hormonal and socio-environmental factors” submitted to the Animals
General comments
I appreciate the opportunity to review this manuscript. This is an investigation on the sociality of male and female cats, considering the age of coexistence with humans and hormonal profile. The test for cat sociality is ingenious, yielding interesting results. However, there are some parts of the text that suggest a limited scope of the experiment. For example, the sample size is limited and the discussion of results is inconclusive. The discussion is incomplete. There are other parts of the text that I suggest improving. Below I explain in detail my doubts and suggestions.
Table S2
In the column sec / time / point” what is defined as "time", it seems more appropriate to define it as frequency.
Methods
What were the observation method and the behavioral recording method?
What type of Correlation analysis is used?
Discussion
The discussion seems fragile and very speculative. The authors argue that neutered male cats have low levels of testosterone, as would be expected in animals without gonads. The "male brain" is formed during fetal development and receives social and environmental inputs during youth and adulthood that reinforce certain behaviors. The authors did not determine how long the males are neutered, and could not determine whether the commitment of male cats was already established for greater sociality before neutering. Also, if testosterone is crucial for a more sociable behavior, why aren't females who have fewer contractions equally sociable as males? In addition, several studies have shown that sociality is influenced by hormones and neurotransmitters such as 5-HT and oxytocin. When the authors discuss masculinization or feminization of the brains of cats, references 23 to 27 are related to experiments on rats. Therefore, it cannot be solidly stated that the same occurs in cats, although it is possible. Thus, one cannot claim that testosterone is intrinsically linked to sociality with the methods used in this behavioral experiment.
The age of sheltering as a marker of the beginning of socialization with humans is an operational definition that is difficult to accept because it hides the entire life history of the animal.
There are other results that are not discussed. There is no discussion about the results in females. The non-difference in hormone concentrations of cortisol and oxytocin is also not discussed.
Many arguments in the discussion do not have a referential basis. It is necessary to establish which articles support the authors' arguments.
In the five paragraphs of the discussion, the authors end in four paragraphs arguing that only the results of the study will be understood in future scientific investigations. Although it is understandable that future studies will bring new understandings of what was observed in the present, it is not desirable to postpone explanations. The authors' writing strategy of postponing the discussion in solid terms weakens the discussion and does not discuss the findings in depth.
Please, note that the sociality of cats toward humans is influenced by both, environmental and inherited factors.
Conclusion
Considering that the discussion is weak, the conclusion is a generalization that is not supported by the data and the experimental design. I suggest rewriting the discussion and conclusion.
Author Response
General comments
I appreciate the opportunity to review this manuscript. This is an investigation on the sociality of male and female cats, considering the age of coexistence with humans and hormonal profile. The test for cat sociality is ingenious, yielding interesting results. However, there are some parts of the text that suggest a limited scope of the experiment. For example, the sample size is limited and the discussion of results is inconclusive. The discussion is incomplete. There are other parts of the text that I suggest improving. Below I explain in detail my doubts and suggestions.
>We appreciate the helpful comments and taking the time to review. Our responses to each comment can be found below.
Table S2
In the column sec / time / point” what is defined as "time", it seems more appropriate to define it as frequency.
>Thank you for your suggestion. We replaced "time" with "frequency".
Methods
What were the observation method and the behavioral recording method?
>We added that the method of observing behavior during the reunion is continuous sampling (line 112-113).
What type of Correlation analysis is used?
>Spearman's rank correlation coefficient was used. We have added the sentence about this (line 182-183).
Discussion
The discussion seems fragile and very speculative. The authors argue that neutered male cats have low levels of testosterone, as would be expected in animals without gonads. The "male brain" is formed during fetal development and receives social and environmental inputs during youth and adulthood that reinforce certain behaviors. The authors did not determine how long the males are neutered, and could not determine whether the commitment of male cats was already established for greater sociality before neutering. Also, if testosterone is crucial for a more sociable behavior, why aren't females who have fewer contractions equally sociable as males? In addition, several studies have shown that sociality is influenced by hormones and neurotransmitters such as 5-HT and oxytocin. When the authors discuss masculinization or feminization of the brains of cats, references 23 to 27 are related to experiments on rats. Therefore, it cannot be solidly stated that the same occurs in cats, although it is possible. Thus, one cannot claim that testosterone is intrinsically linked to sociality with the methods used in this behavioral experiment.
>We thank the reviewer for this comment. As the reviewer says, we do not know when castration occurred. The period of sexual differentiation of brain in cats has not been known yet, but in most mammals it occurs around birth (McCarthy et al., 2018). We assumed that the cats in this experiment were not castrated before the sexual differentiation period of the brain, as they could not be anesthetized around birth. To improve clarity, the discussion about testosterone and behaviors has been modified (line 222-235).
Serotonin and oxytocin also influence cat behavior (Cools 1973; Koyasu et al., 2022), but relationships with testosterone are not clear. We have added the relationships between hormones (line 262-266).
In addition, our data do not suggest that females are unsociable, but rather that males possibly have testosterone-dependent behaviors toward humans due to that male cat brain can be masculinized during the perinatal period and show a different response to testosterone in adulthood as compared to females.
The age of sheltering as a marker of the beginning of socialization with humans is an operational definition that is difficult to accept because it hides the entire life history of the animal.
>We have added to the limitation paragraph that there is no certainty that the age in months when they were rescued is when they began interacting with humans (line 282-287).
There are other results that are not discussed. There is no discussion about the results in females. The non-difference in hormone concentrations of cortisol and oxytocin is also not discussed.
>It was possible that the lack of a relationship between testosterone and behavior in females was due to sexual differentiation of the brain, which may not have resulted in testosterone-dependent behaviors. The discussion has been organized and modified (line 222-235). In addition, we have added a discussion of the lack of correlation between cortisol, oxytocin and behaviors (line 245-261).
Many arguments in the discussion do not have a referential basis. It is necessary to establish which articles support the authors' arguments.
>References have been added in discussion as needed .
・Effects of chronically high doses of the anabolic androgenic steroid, testosterone, on intermale aggression and sexual behavior in male rats, Lumia AR, Thorner KM, McGinnis MY, Physiol. Behav., 1994
・Intermale aggression in mice: does hour of castration after birth influence adult behavior?, Motelica-Heino I, Edwards DA, Roffi J, Physiol. Behav., 1993
・Anabolic-androgenic steroid exposure during adolescence and aggressive behavior in golden hamsters, Melloni Jr RH, Connor DF, Hang PT, Harrison RJ, Ferris CF, Physiol. Behav., 1997
・Excessive mortality in young free-ranging male nonhuman primates with low cerebrospinal fluid 5-hydroxyindoleacetic acid concentrations, Higley JD, Mehlman PT, Higley SB, Fernald B, Vickers J, Lindell SG et al., Arch. Gen. Psychiatry, 1996
・CSF 5-HIAA, testosterone, and sociosexual behaviors in free-ranging male rhesus macaques in the mating season, Mehlman PT, Higley JD, Fernald BJ, Sallee FR, Suomi SJ, Linnoila M, Psychiatry Res., 1997
・Primate models to understand human aggression, Kalin NH, J. Clin. Psychiatry, 1999
・Anabolic steroids and aggressive behavior in cynomolgus monkeys, Rejeski WJ, Brubaker PH, Herb RA, Kaplan JR, Koritnik D, J. Behav. Med., 1988
and 15 other references
In the five paragraphs of the discussion, the authors end in four paragraphs arguing that only the results of the study will be understood in future scientific investigations. Although it is understandable that future studies will bring new understandings of what was observed in the present, it is not desirable to postpone explanations. The authors' writing strategy of postponing the discussion in solid terms weakens the discussion and does not discuss the findings in depth.
>Thank you for your comment. A paragraph on the limitations of the study has been included, and each paragraph describes only what is considered possible from the results.
Please, note that the sociality of cats toward humans is influenced by both, environmental and inherited factors.
>Corrected the word from genetic to inherited (line 16, 274).
Conclusion
Considering that the discussion is weak, the conclusion is a generalization that is not supported by the data and the experimental design. I suggest rewriting the discussion and conclusion.
>Discussion and conclusions were revised based on reviewer 2's comments.
Round 2
Reviewer 1 Report
In general, the edits have improved the ms.
Reference 10 should refer to Fogle (spelling)
It could be worth mentioning that neutered males may be generally friendlier than neutered females (Hart & Hart, Your Ideal Cat)
Author Response
>We are thankful for your time and energy.
In general, the edits have improved the ms.
Reference 10 should refer to Fogle (spelling)
> The spelling has been corrected.
It could be worth mentioning that neutered males may be generally friendlier than neutered females (Hart & Hart, Your Ideal Cat)
> Thank you for the suggestion. We have added the following sentences (line 328-331): “In addition, it has been reported that neutered males are friendlier to humans than neutered females [40]. The sex differences in hormone-behavior associations shown in this study may help us understand sex differences in behavior toward humans.”
Reviewer 2 Report
Dear Editor
Thank you for the opportunity to review the manuscript again. I read the author's response. Most of the suggestions were accepted in the first review round. Overall, the revised version has improved and appears to have achieved sufficient quality and clarity to be published.
A small but important suggestion: I suggest changing the term "periphery" to "peripheral tissues" in line 233.
Author Response
Thank you for the opportunity to review the manuscript again. I read the author's response. Most of the suggestions were accepted in the first review round. Overall, the revised version has improved and appears to have achieved sufficient quality and clarity to be published.
>We are thankful for your time and energy.
A small but important suggestion: I suggest changing the term "periphery" to "peripheral tissues" in line 233.
>Thank you for pointing this out. According to the comment, we have corrected it (line 326).